# *Thevetia thevetioides* Cardenolide and Related Cardiac Glycoside Profile in Mature and Immature Seeds by High-Resolution Thin-Layer Chromatography (HPTLC) and Quadrupole Time of Flight–Tandem Mass Spectrometry (Q-TOF MS/MS) Reveals Insights of the Cardenolide Biosynthetic Pathway

**DOI:** 10.3390/molecules29174083

**Published:** 2024-08-28

**Authors:** Juan Vázquez-Martínez, Paulina Bravo-Villa, Jorge Molina-Torres

**Affiliations:** 1Departamento de Ingeniería Química, Tecnológico Nacional de México/ITS Irapuato, Silao-Irapuato km 12.5 El Copal, Irapuato 36821, GTO, Mexico; juan.vm@irapuato.tecnm.mx; 2Departamento de Biotecnología y Bioquímica, Centro de Investigación y de Estudios Avanzados del IPN Unidad Irapuato, Libramiento Norte Carretera Irapuato León Kilómetro 9.6, Carr Panamericana Irapuato León, Irapuato 36821, GTO, Mexico

**Keywords:** *Thevetia thevetioides*, *Thevetia peruviana*, cardiac glycosides, HPTLC, biosynthetic pathways, Na^+^/K^+^ ATPase

## Abstract

*Thevetia thevetioides* is a species within the Apocynaceae family known for containing cardenolide-glycosides, commonly referred to as cardiac glycosides, which are characteristic of this genus. The seeds of the *Thevetia* species are frequently used as a model source for studying cardiac steroids, as these glycosides can be more readily extracted from the oil-rich seeds than from the plant’s green tissues. In this work, the cardenolide profile of ripe and immature seeds was determined and compared to establish the main differences. Ripe seeds contain six related cardenolides and triosides, with thevetin B being the predominant component. In contrast, immature seeds exhibit a total of thirteen cardiac glycosides, including monoglycosides such as neriifolin and peruvosides A, B, and C, as well as diglycosides like thevebiosides A, B, and C. Some of these compounds have previously been identified as degradation products of more complex cardiac glycosides; however, their presence in immature seeds, as described in this study, suggests that they may serve as biosynthetic precursors to the triosides observed in mature seeds. The glycoside patterns observed via HPTLC are associated with specific chemical structures characteristic of this genus, typically featuring thevetose or acetyl-thevetose at the first position, followed by glucose or gentibiose in di- or trisaccharides, independent of the trioside aglycones identified: digitoxigenin, cannogenin, or yccotligenin. Ripe seeds predominantly contain triosides, including thevetin B, C, and A, the latter of which has not been previously reported.

## 1. Introduction

Active compounds identified and isolated from plants have the potential to be developed into drugs with enhanced efficacy and/or reduced toxicity. This study focuses on cardenolides, also known as cardiac glycosides, which are present in several plant species. However, these compounds have been observed to exert toxic effects in humans and other animals [1,2]. Both intentional and accidental exposure to these compounds can adversely affect the cardiovascular system, leading to serious health issues and even death [1]. The effects of cardenolides include arrhythmias, sinus bradycardia, and other disturbances caused by the inhibition of the Na^+^/K^+^ ATPase system [2,3,4,5,6]. Cardiac glycosides are steroidal glycosides that specifically influence the dynamics and rhythm of the heart muscle [4,6]. These glycosides bind to and inactivate the Na^+^/K^+^ ATPase pump on the cytoplasmic membrane of cardiac cells [7]. Structurally, cardenolic steroids are derived from the 10,13-dimethylperhydrocyclopenta[*a*]phenanthrene tetracyclic ring system and are characterized by the presence of a γ-lactone ring (cardenolides) or a δ-lactone ring (bufadienolides) attached at the β position at C17. Typical sugar residues in these compounds are derived from deoxy and/or C3-O-methylated sugars, which are glycosidically linked through the C3-OH group of the steroid backbone [1,2,3,4,5]. The absence of an unsaturated lactone ring renders the glycosides cardio-inactive [3,6].

*Thevetia thevetioides* (HBK) K. Schum (synonyms: *Cervera thevetioides* HBK in Nov. Gen. Sp. 3: 223, 1819; *Thevetia yccotly* A. DC. in DC. Prodr. 8: 343, 1844; *Thevetia yccotly glabra* A. DC. in DC. Prodr. 8: 343, 1844) belongs to the Apocynaceae family [8]. This species is endemic to Mesoamerica, with a distribution ranging from central and southern Mexico to Central and South America. It is a visually striking tree, notable for its fruits, foliage, and flowers, which make it more attractive than other species in its genus. Although it is cultivated ornamentally, it is less commonly grown than *Thevetia peruviana* [3,9]. Its Nahuatl name, *yoyotli*, means sleighbell (cascabel). The vegetative parts of *T. thevetioides* have recognized uses in traditional medicine, including treatments for conditions such as hemorrhoids; the milky sap of the leaves is used to treat deafness, ulcers, and scabies; and the leaves are applied to soothe sore teeth and treat tumors [10,11,12]. In general, species within this genus are considered toxic. Although it has been taxonomically misassociated with *Nerium oleander*, *T. thevetioides* is also known for its potent cardiac glycosides [5,10,13].

Numerous phytochemical studies have demonstrated that the *Thevetia* genus, particularly its seeds, is rich in cardenolides. The therapeutic and toxic properties of this plant have been recognized for a long time, dating back to 1863 when De Vry isolated and named thevetin from the defatted seeds [13]. Helfenberger and Reichstein later discovered that *Thevetia* seeds are rich in enzymes capable of hydrolytically eliminating two glucose residues. When defatted seed powder was incubated with water for several days, neriifolin was produced in significant amounts [14]. The enzymatic hydrolysis, which separates the two glycosides, was further detailed, revealing that neriifolin represents the glucose-free portion of thevetin. Thevetin cleaves into neriifolin and two moles of D-glucose. Under milder enzymatic conditions, thevebioside, containing one mole of glucose, was obtained; it subsequently decomposes into neriifolin and glucose upon repeated enzymatic treatment. Therefore, both neriifolin and thevetin have digitoxigenin as their aglycone base. Various compounds, including neriifolin, acetyl-neriifolin, isolated iridoids, terpenoids, alkaloids, flavonoids, saponins, and tannins, have been isolated from *T. thevetioides* [14]. These observations led to the suggestion that mono- and diglycosides are generated as degradation products of more complex cardiac glycosides.

For the elucidation of the triose structure [β-D-glucopyranosyl-(1→6)-β-D-glucopyranosyl-(1→4)-6-deoxy-3-O-methyl-α-L-glucopyranosyl, or D-Glc-D-Glc-L-Thev], acid hydrolysis was performed, yielding gentibiose (diglucose). A similar approach was used for the characterization of cannogenin, thevetose (L-Thev), and thevetin acetates. In 1945, Frèrejacque described the sugar L-Thev (6-deoxy-3-O-methyl-L-glucopyranoside) as a hexose-methyl ether [15]. A wide variety of sugars can bind to natural cardiac glycosides, with the most common being glucose, galactose, mannose, rhamnose, and digitalose. Although the sugars themselves do not contribute to biological activity, they serve as important taxonomic markers. Specifically, the genus *Thevetia* is characterized by L-Thev as the initial sugar glycoside, followed by two glucose units. Additionally, the presence of acetylation in L-Thev (L-ThevAc) is observed in the tri-, di-, and monoglycosides of *Thevetia* species (Figure 1) [16,17].

Aglycones can be obtained by cleaving the sugar moieties through acid hydrolysis or enzymatic methods. In hydrolysates of *Thevetia thevetioides* seeds, three aglycones have been characterized: cannogenin, digitoxigenin, and (3β,5β,20R)-18,20-epoxy-3,14-dihydroxycardanolide (here referred to as Yccotligenin) (Figure 1) [6]. Most previous studies on *T. thevetioides* seeds have focused on mature seeds. The objective of this work is to characterize the cardenolides present in both immature and dry mature seeds of *Thevetia thevetioides*, comparing their profiles using high-resolution thin-layer chromatography (HPTLC) and analyzing the main components of the HPTLC chromatograms by Q-TOF MS.

## 2. Materials and Methods

### 2.1. Plant Material

*Thevetia thevetioides* seeds were collected from Rancho Machi, Santa Cruz de Juventino Rosas (20°35′06.99″ N, 100°57′33.94″ W, 1741 m.a.s.l.), Guanajuato, Mexico. Seeds were gathered at both immature (characterized by green pericarp, latex, and soft testa) and mature stages (naturally dried fruits). In the immature stage, the seed embryo consists of a gelatinous material. In contrast, mature fruits exhibit a dark green pericarp and a hard testa. The fruit structure includes an embryonic axis with four cotyledons. The fruits were transported on ice and immediately processed in the laboratory as described below.

### 2.2. Extraction

The mature and immature seeds were separated from the pericarp, leaving only the embryo and cotyledons. These were then weighed and homogenized. Replicates of 1 g each were placed in thimbles for an initial extraction using 80 mL of hexane to degrease the seeds, utilizing a Soxhlet Buchi Extraction Unit E-816 ECE (BÜCHI Labortechnik AG, Flawil, Switzerland). The extraction was performed at a plate temperature of 80 °C for 6 cycles (10 min per cycle). This phase, primarily containing lipids (triglycerides), was discarded. Subsequently, the seeds were extracted with 90% ethanol to isolate the cardiac glycosides, again at a plate temperature of 80 °C for 6 cycles (12 min per cycle). The ethanolic extract was then concentrated using a Multivapor™ P-6/P-12 with a Vacuum Pump V-100 oil-free vacuum pump, controlled by a Büchi Interface I-100 temperature and pressure control board (BÜCHI Labortechnik AG, Flawil, Switzerland). Finally, the residue containing the cardiac glycosides was resuspended in 90% ethanol [18,19].

### 2.3. HPTLC

The plates used were HPTLC 20 × 10 cm (HPTLC Fertigplatten Nano-Adamant UV254) from Macherey & Nagel, Düren, Germany. Sample application was performed using an Automatic TLC Sampler 4 (ATS4) from CAMAG, Muttenz, Switzerland. The application conditions on the ATS4 were as follows: application gas: N_2_, application rate: 150 nL/s, type of application: bands, and band length: 6 mm.

After sample application, the plate was developed in an Automatic Developing Chamber 2 (ADC 2, CAMAG, Muttenz, Switzerland) using a mobile phase of ethyl acetate: methanol: water (81:11:8 *v*/*v*). The development conditions were as follows: exposure time to relative humidity: 5 min, saturation time: 10 min, saturation fill volume: 25 mL, development fill volume: 10 mL, and solvent front migration distance: 80 mm. The plate was dried for 5 min after development. Cardiotonic glycosides were detected using Kedde reagent, which reacts with the γ-lactone ring to produce a coloration ranging from pink to blue-violet under white light. This coloration fades within minutes but can be restored with additional reagent application. Chromatogram images were captured using the TLC Visualizer from CAMAG (Muttenz, Switzerland) [18,19].

The system for application, development, visualization, and quantification was managed using the visionCATS software version 2.0, which organizes the HPTLC analysis workflow, controls CAMAG instruments, and facilitates data evaluation.

### 2.4. Q-TOF MS/MS

To identify the cardenolides present in the ethanolic extracts from mature and immature *T. thevetioides* seeds by mass spectrometry, 100 µL of each extract was injected manually. The mass spectra were obtained using a Q-TOF mass spectrometer with orthogonal geometry (Micromass, Wilmslow, UK), equipped with a nanospray ion source. The borosilicate capillary and cone voltages were set at 900 and 35 volts, respectively. Argon was used as the collision gas, and the collision energy was varied between 15 and 50 eV to obtain mass spectra with sufficient structural information for reliable identification of the analytes in the databases. The mass spectra were processed using MassLynx version 4.0 software (Micromass, Wilmslow, UK) [16]. For detailed methodology, see [6].

### 2.5. GC-MS Analysis of Aglycones

To confirm the identity of the aglycones present in the glycosides of *T. thevetioides*, the glycosides were hydrolyzed as follows: 100 µL of the extract from immature seeds was placed in a reactivial (Supelco, Inc., Bellefonte, PA, USA) and evaporated to dryness under a nitrogen stream. Next, 1 mL of 10 M HCl was added to the sample, which was then sealed and incubated at 70 °C for 2 h. After cooling to room temperature, the excess HCl was evaporated under nitrogen. The hydrolysate was then derivatized by adding 80 µL of BSTFA + 1% TMC and 20 µL of pyridine, and the mixture was incubated at 80 °C for 30 min. The derivatized sample was analyzed using GC-MS, with parameters adjusted to achieve optimal resolution for each aglycone. Compound identification was performed by comparing the mass spectra to those in the NIST Library and with previously reported spectra. For detailed methodology, see [6].

## 3. Results

The identification of the cardenolides was performed according to the method described by Kohls [6]. Compounds were classified based on the number and type of sugar moieties and the aglycone present. The cardenolides were categorized into groups: triglycosides, acetylated triglycosides, diglycosides, acetylated monoglycosides, and monoglycosides. GC-MS analysis allowed for the identification of three aglycones: cannogenin, yccotligenin [(3β,5β,20R)-18,20-epoxy-3,14-dihydroxycardanolide], and digitoxigenin.

### 3.1. Mature Seed Cardenolide Profile

The extract from mature *T. thevetioides* seeds, separated on HPTLC plates, exhibits a relatively simple cardenolide profile (Figure 2 and Table 1). It primarily contains thevetin A, which consists of the aglycone cannogenin (C15) attached to the monosaccharide L-Thev and two glucose residues. Thevetin B and thevetin C differ from thevetin A only in their aglycones; thevetin B has digitoxigenin (C13) and thevetin C has yccotligenin (C14). The acetylated forms, thevetin A acetate, thevetin B acetate, and thevetin C acetate, share the same aglycones as their non-acetylated counterparts, with the distinction that L-Thev is acetylated in each case. Additionally, thevebiosides A (C9) and C (C8) were detected at low concentrations according to the TLC profile. The structures of the aglycones and sugars are illustrated in Figure 1. See Appendix A for the MS/MS analysis of the compounds.

### 3.2. Immature Seeds Cardenolide Profile

Immature *T. thevetioides* seeds exhibited a significantly higher concentration and diversity of cardiac glycosides as evidenced by HPTLC (Figure 2 and Table 1), including monoglycosides such as peruvoside A (C6), which comprises the aglycone cannogenin and the monosaccharide L-Thev; peruvoside B (C4), which contains the aglycone digitoxigenin and L-Thev; and peruvoside C (C5), which includes the aglycone yccotligenin and L-Thev. Additionally, immature seeds contain the acetylated forms of these peruvosides: peruvoside A acetate, peruvoside B acetate, and peruvoside C acetate, where the respective aglycone is bound to one molecule of acetylated thevetose (L-ThevAc) (C3, C1, C2, respectively). The structures of the aglycones and sugars are shown in Figure 1. See Appendix A for the MS/MS analysis of the compounds.

Furthermore, immature seeds also contain significant amounts of the diglycosides thevebiosides A, B, and C. These compounds have the aglycones cannogenin, digitoxigenin, and yccotligenin, respectively, with each containing a disaccharide consisting of one glucose and one L-Thev linked to the aglycone. The thevetins A, B, and C and their acetylated forms, thevetin A acetate, thevetin B acetate, and thevetin C acetate were also identified. The structures of the aglycones and sugars are illustrated in Figure 1. See Appendix A for the MS/MS analysis of the compounds.

## 4. Discussion

The solvent system used was effective for resolving the bands and staining the glycosides as described by Wagner and Bladt [18], facilitating the comparison between mature and immature seeds. High-resolution thin-layer chromatography (HPTLC) offers flexibility in sample quantity and application methods, ranging from manual, simple, and cost-effective techniques to sophisticated and instrumentally standardized approaches. Electrospray ionization combined with mass spectrometry proved efficient for analyzing the non-volatile and polar cardiac glycosides extracted from the HPTLC plate silica [20]. During Q-TOF MS/MS analysis, ammonium adducts of *T. thevetioides* glycosides were observed, formed from ammonium ions in the mobile phase, as well as sodium adducts due to the presence of sodium ions; the latter were considered in the compound identification (See Appendix A) [20]. Our results indicate that complex glycosides, such as diglycosides and triglycosides, are synthesized from monoglycosides, contrary to previous beliefs. Thus, the mechanisms of sugar condensation and acetylation require further investigation. Although the aglycone of cardenolides is crucial for determining bioactivity, it is also essential to characterize the bioactivity of each glycoside in *T. thevetioides* to explore potential applications and understand the role of sugar modifications in their biological activity [21]. We propose renaming some glycosides, as outlined in Table 1, to improve structural identification and understanding.

During HPTLC analysis, the migration of glycosides is correlated with the number of sugar moieties. Monoglycosides typically migrate to an Rf value of between 0.5 and 0.9, diglycosides to an Rf value of between 0.2 and 0.4, and triglycosides near the application zone to an Rf value of less than 0.2. Comparison of cardenolides with the same sugar composition but different aglycones reveals that cannogenin is the most polar compound, as it is strongly retained by the stationary phase, whereas digitoxigenin is the least polar, as it migrates further with the mobile phase. Yccotligenin exhibits intermediate polarity. Consequently, the bioactivity of these glycosides may be related to their polarity and corresponding water solubility [17,19,20].

## 5. Conclusions

The integration of HPTLC and Q-TOF MS/MS represents a robust approach for characterizing the non-volatile components of plant extracts. In this study, the cardenolide profiles of mature and immature *T. thevetioides* seeds were analyzed, revealing distinct compositions for each developmental stage. Comparative analysis of these profiles supports the hypothesis that complex cardenolides are synthesized from monoglycosides rather than being derived from their degradation. Further research is necessary to elucidate the mechanisms of cardenolide synthesis and to explore the bioactivity of *T. thevetioides* cardenolides.

## Figures and Tables

**Figure 1 molecules-29-04083-f001:**
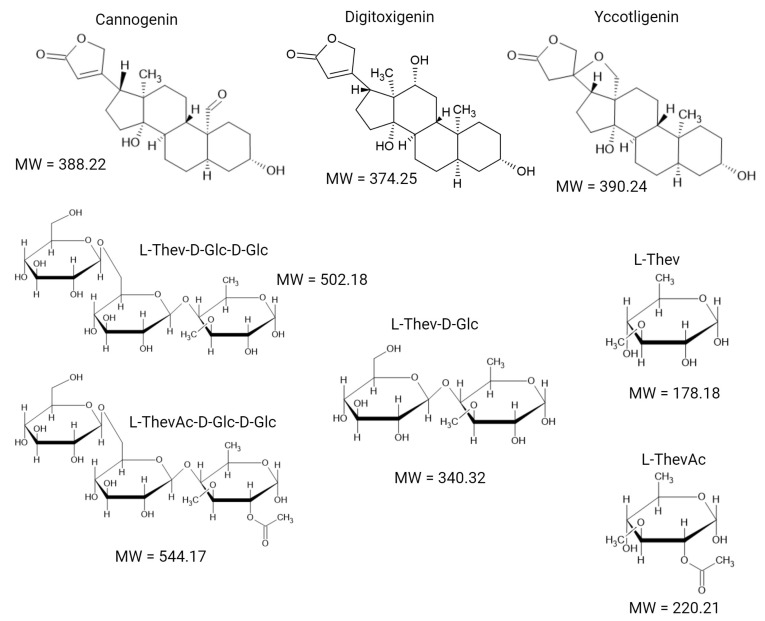
Chemical structures of characteristic aglycones and sugars of *T. thevetioides* cardenolides.

**Figure 2 molecules-29-04083-f002:**
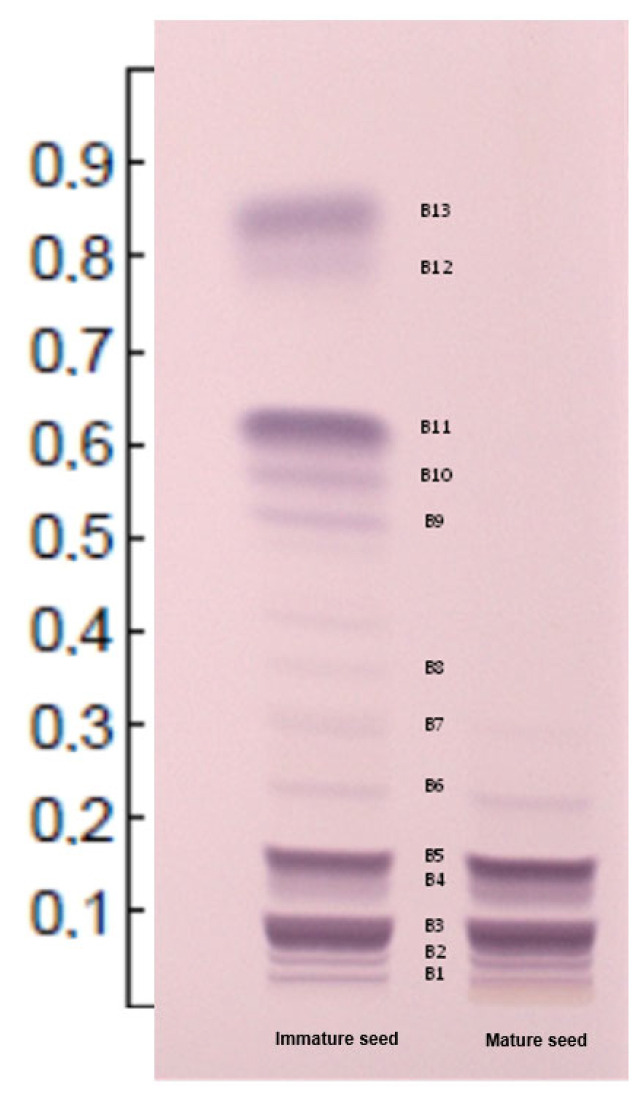
High-resolution thin-layer chromatography (HPTLC) profile comparing mature and immature seeds of *T. thevetioides*. See Appendix A for complementary information.

**Table 1 molecules-29-04083-t001:** Cardiac glycosides identified in mature and immature seeds of *Thevetia thevetioides*. The compounds correspond to the bands shown in Figure 1, as separated by HPTLC. Identification was performed using Q-TOF MS/MS. Due to the diverse and variable nomenclature of these components, we propose a standardized naming convention: first indicating the aglycone and then the species from which it was originally isolated. “I” denotes immature seeds and “M” denotes mature seeds. See Appendix A for complementary information.

ID	Seed	BAND	Rf	Aglycone	Carbohydrate	Previous Name	Proposed Name		
**C1**	**I**	**B13**	0.84	Digitoxigenin	L-ThevAc	Peruvoside B acetate/Neriifolin acetate (Cerberin)	Digiperuvoside acetate or Digithevetoside acetate	**Monoglycosides**	**Acetates**
**C2**	**I**	**B12**	0.78	Yccotligenin	L-ThevAc	Peruvoside C acetate	Yccoperuvoside acetate or Yccothevetoside acetate		
**C3**	**I**			Cannogenin	L-ThevAc	Peruvoside A acetate	Canperuvoside acetate or Canthevetoside acetate		
**C4**	**I**	**B11**	0.61	Digitoxigenin	L-L-Thev	Peruvoside B (Neriifolin)	Digiperuvoside or Digithevetoside		
**C5**	**I**	**B10**	0.56	Yccotligenin	L-Thev	Peruvoside C	Yccoperuvoside or Yccothevetoside		
**C6**	**I**	**B9**	0.51	Cannogenin	L-Thev	Peruvoside A	Canperuvoside or Canthevetoside		
**C7**	**I**	**B8**	0.35	Digitoxigenin	L-Thev-D-Glc	Thevebioside B	Digithevebioside	**Di-glycosides**	
**C8**	**M/I**	**B7**	0.29	Yccotligenin	L-Thev-D-Glc	Thevebioside C	Yccothevebioside		
**C9**	**M/I**	**B6**	0.22	Cannogenin	L-Thev-D-Glc	Thevebioside A	Canthevebioside		
**C10**	**M/I**	**B5**	0.15	Digitoxigenin	L-ThevAc-D-Glc-D-Glc	Thevetin B acetate	Digithevetin acetate	**Triglycosides**	**Acetates**
**C11**	**M/I**	**B4**	0.12	Yccotligenin	L-ThevAc-D-Glc-D-Glc	Thevetin C acetate	Yccothevetin acetate		
**C12**	**M/I**			Cannogenin	L-ThevAc-D-Glc-D-Glc	Thevetin A acetate	Canthevetin acetate		
**C13**	**M/I**	**B3**	0.07	Digitoxigenin	L-Thev-D-Glc-D-Glc	Thevetin B (Cerberoside)	Digithevetin		
**C14**	**M/I**	**B2**	0.04	Yccotligenin	L-Thev-D-Glc-D-Glc	Thevetin C	Yccothevetin		
**C15**	**M/I**	**B1**	0.02	Cannogenin	L-Thev-D-Glc-D-Glc	Thevetin A	Canthevetin		

## Data Availability

The data will be shared under demand via email.

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
