# Peer review of "Thevetia thevetioides Cardenolide and Related Cardiac Glycoside Profile in Mature and Immature Seeds by High-Resolution Thin-Layer Chromatography (HPTLC) and Quadrupole Time of Flight–Tandem Mass Spectrometry (Q-TOF MS/MS) Reveals Insights of the Cardenolide Biosynthetic Pathway"

_molecules, 2024, doi:10.3390/molecules29174083_

Round 1

Reviewer 1 Report

Comments and Suggestions for Authors

It would be appropriate to mention at a suitable place that the trisaccharide group in the compounds under discussion is β-d-glucopyranosyl-(1→6)-β- d -glucopyranosyl-(1→4)-6-deoxy-3-O-methyl-α-l-glucopyranosyl, for 6-deoxy-3-O-methyl-l-glucopyranose the name l-thevetose is used, and the abbreviation l-Thev or just Thev is used for it. For acetylated l-thevetose, the symbol ThevAc could be introduced.

Lines 36-37: The part of the sentence "consist … glycoside" is messy and unnecessary.

Lines 39-40: Let “10,13-dimethyl-cyclopentaneperhydrophenanthrene“ be corrected to “10,13-dimethylperhydrocyclopenta[a]phenanthrene“.

Lines 79-81: The text should be improved stylistically and in terms of language.

Figure 1: In structural formulae of saccharides, the configuration on all asymmetric atoms should be expressed using bold wedge-shaped and cross-hatched wedge-shaped bonds.

Lines 137, 148, 152 and 153: "uL" is to be replaced by "μL".

Compound names, when they are not protected (commercial) names and when they are not at the beginning of a sentence, there is no reason to write with a capital letter.

There are significant discrepancies in Appendix A. Ions should be expressed in a standard way with the sign of electric charge in the superscript after the square bracket. In the composition of ions, it is necessary to distinguish the signs - (hyphen) for the bond and – (minus). Oligosaccharide compositions should be reported in accordance with IUPAC and IUBMC conventions, i.e. for example "l-Thev-d-Glc- d -Glc" or simply "Thev-Glc-Glc". It should be explained in advance that, for example, "M – Glc" simply expresses the product of the hydrolytic cleavage of the glucose molecule from the M molecule, i.e. exactly "M + H2O – Glc", "M – AcOH" simply expresses the product of the hydrolytic cleavage of acetic acid molecule from the M molecule, i.e. exactly "M + H2O – AcOH". It would not be correct, for example,  to write "[M – Glc + Na]+ = 733.26", but the way of writing "[M – Glc + Na]+, m/z = 733.26" could be suitable.

Examples of inconsistences in the first few paragraphs are as follows:

Lines 248, 254, 260, and 270: the stated value of m/z [l-Thev-d-Glc-d-Glc + Na]+ 507.09 contradicts the calculated value 525.18.

Line 250: The value of m/z [l-Thev-d-Glc + Na]+ 345.09 contradicts the calculated value 363.13.

Lines 262 and 266: "Thevetina" (3 occurrences) is to be corrected to "thevetin".

Line 263: The thevetin C aglycone is not thevetiogenin (for which the manuscript proposes the name Yccotligenin) but (3β,5β,20R)-18,20-epoxy-3,14-dihydroxycardanolide (CAS RN 71129-77-6). The value of m/z of thevetin C on line 252, as well as the values  of m/z of the products of its partial hydrolysis ([M – Glc + Na]+ and [M – Glc-Glc + Na]+) on lines 253 and 254 are correct.

Line 264: Let "cannogenin" be corrected to "canogenin".

Lines 265-266: the wording "first Thevetose of both" needs to be corrected to "thevetose unit of the trisaccharide chain" for example.

Row 267: A matching ion designation could be [M – AcOH + Na]+.

Line 270: A matching ion designation could be [ThevAc-Glc-Glc – AcOH + Na]+.

Line 272: Correct is "gentiobiose" or "cellobiose" instead of "Gentobiose".

It is not only the given examples that need to be dealt with, but the entire section Appendix A is necessary to be checked and corrected and, in addition, Table 1 is necessary to be brought into line.

In References, it would be appropriate to use abbreviation titles of journals according to CAS Source Index: [2] Toxin Rev.; [3] Auton. Autacoid Pharmacol.; [4,12] Phytochemistry (Elsevier); [13] Rev. Bras. Farmacogn.; [15] Ann. Bogor.; [16] Food Addit. Contam.: Part A.

The journal in [9] has numbered pages in each issue separately, so it is necessary to include the issue: Int. J. Pharm. Sci. Invent. 2014, 3(4), pp: 11-16.

Line 345: "Thevetin. Helv Chim Acta" be fixed to "Thevetin. I. Glycoside und Aglykone. Helv. Chim. Acta"

In [11] the given title of the article as well as the volume and pages are incorrect. The correct wording of the entire reference is: [11] Frèrejacque, M. La nériifoline, nouvel hétéroside digitalique de Thevetia neriifolia. C. R. Hebd. Seances Acad. Sci. 1945, 221, pp. 645-646.

Author Response

Reviewer 1

It would be appropriate to mention at a suitable place that the trisaccharide group in the compounds under discussion is β-d-glucopyranosyl-(1→6)-β- d -glucopyranosyl-(1→4)-6-deoxy-3-O-methyl-α-l-glucopyranosyl, for 6-deoxy-3-O-methyl-l-glucopyranose the name l-thevetose is used, and the abbreviation l-Thev or just Thev is used for it. For acetylated l-thevetose, the symbol ThevAc could be introduced.

R= Addressed

Lines 36-37: The part of the sentence "consist … glycoside" is messy and unnecessary.

R= Addressed Lines 39-40: Let “10,13-dimethyl-cyclopentaneperhydrophenanthrene“ be corrected to “10,13-dimethylperhydrocyclopenta[a]phenanthrene“.

R= Addressed

Lines 79-81: The text should be improved stylistically and in terms of language.

R=Addressed

Figure 1: In structural formulae of saccharides, the configuration on all asymmetric atoms should be expressed using bold wedge-shaped and cross-hatched wedge-shaped bonds.

R=Addressed

Lines 137, 148, 152 and 153: "uL" is to be replaced by "μL".

R= Addressed

Compound names, when they are not protected (commercial) names and when they are not at the beginning of a sentence, there is no reason to write with a capital letter.

R=Addressed

There are significant discrepancies in Appendix A. Ions should be expressed in a standard way with the sign of electric charge in the superscript after the square bracket. In the composition of ions, it is necessary to distinguish the signs - (hyphen) for the bond and – (minus). Oligosaccharide compositions should be reported in accordance with IUPAC and IUBMC conventions, i.e. for example "l-Thev-d-Glc- d -Glc" or simply "Thev-Glc-Glc". It should be explained in advance that, for example, "M – Glc" simply expresses the product of the hydrolytic cleavage of the glucose molecule from the M molecule, i.e. exactly "M + H2O – Glc", "M – AcOH" simply expresses the product of the hydrolytic cleavage of acetic acid molecule from the M molecule, i.e. exactly "M + H2O – AcOH". It would not be correct, for example,  to write "[M – Glc + Na]+ = 733.26", but the way of writing "[M – Glc + Na]+m/z = 733.26" could be suitable.

R=All the Appendix A was corrected and homogenized.

Examples of inconsistences in the first few paragraphs are as follows:

Lines 248, 254, 260, and 270: the stated value of m/z [l-Thev-d-Glc-d-Glc + Na]+ 507.09 contradicts the calculated value 525.18.

R=After fragmentation, the trisaccharide can cleavage leaving the oxygen linked to the aglycone, giving 485 m/z, so the sodium adduct have 507 m/z approx. This has been previously described by Kohls (2012). For better understanding the [L-Glc-L-Glc-L-Thev + Na – H2O]+ 

Line 250: The value of m/z [l-Thev-d-Glc + Na]+ 345.09 contradicts the calculated value 363.13.

R= Same as above

Lines 262 and 266: "Thevetina" (3 occurrences) is to be corrected to "thevetin".

R= Addressed

Line 263: The thevetin C aglycone is not thevetiogenin (for which the manuscript proposes the name Yccotligenin) but (3β,5β,20R)-18,20-epoxy-3,14-dihydroxycardanolide (CAS RN 71129-77-6). The value of m/z of thevetin C on line 252, as well as the values  of m/z of the products of its partial hydrolysis ([M – Glc + Na]+ and [M – Glc-Glc + Na]+) on lines 253 and 254 are correct.

R=Addressed, text and figure were corrected.

Line 264: Let "cannogenin" be corrected to "canogenin".

R= Addressed, cannogenin was homogenized.

Lines 265-266: the wording "first Thevetose of both" needs to be corrected to "thevetose unit of the trisaccharide chain" for example.

R= Addressed

Row 267: A matching ion designation could be [M – AcOH + Na]+.

R=Addressed

Line 270: A matching ion designation could be [ThevAc-Glc-Glc – AcOH + Na]+.

R=Addressed

Line 272: Correct is "gentiobiose" or "cellobiose" instead of "Gentobiose".

R=Corrected

It is not only the given examples that need to be dealt with, but the entire section Appendix A is necessary to be checked and corrected and, in addition, Table 1 is necessary to be brought into line.

R=It was corrected, when necessary, considering the reviewer comments and the previously described by Kohls (2012).

In References, it would be appropriate to use abbreviation titles of journals according to CAS Source Index: [2] Toxin Rev.; [3] Auton. Autacoid Pharmacol.; [4,12] Phytochemistry (Elsevier); [13] Rev. Bras. Farmacogn.; [15] Ann. Bogor.; [16] Food Addit. Contam.: Part A.

R=Addressed, the dot after abbreviation was not included according to Molecules guidelines.

The journal in [9] has numbered pages in each issue separately, so it is necessary to include the issue: Int. J. Pharm. Sci. Invent. 20143(4), pp: 11-16.

R=Addressed

Line 345: "ThevetinHelv Chim Acta" be fixed to "Thevetin. I. Glycoside und Aglykone. Helv. Chim. Acta"

R=Addressed

In [11] the given title of the article as well as the volume and pages are incorrect. The correct wording of the entire reference is: [11] Frèrejacque, M. La nériifoline, nouvel hétéroside digitalique de Thevetia neriifolia. C. R. Hebd. Seances Acad. Sci. 1945221, pp. 645-646.

R=Addressed

Reviewer 2 Report

Comments and Suggestions for Authors

The paper need to revised.

The introduction should be implemented with more recent papers. The authors describe method very old (1926, 1933, 1945, 1948, 1961)

2) 2.1 Plant material -  line 97 correct "stage. stage"

Describe as stored the sample before analysis. (Temperature)

2.2 extraction - the authors apply an extraction method very time and solvent consuming, a justification should be given on the choice of this method. In the lecterature, there are many other extraction method more appropriate.

2.4 Q-TOF MS/MS - the authors should give more informations. Ion selected for qualify and quantify. A description of the selction of the ions and the parameters of ion detection should be given in table.

2.5 GC-MS analysis Information on the GC condition should be given (column type, oven condition, MS condition) The ions more adbundant used for the qualy-quanti analysis.

References - The authors should implement the reference with more recent papers.

In additon the authors should give information on the recoveries obtained with the applycation of method. (Fortification levels, number of replicates, acuracy, precision, linearity, LOD and LOQ)

Author Response

Reviewer 2

The introduction should be implemented with more recent papers. The authors describe method very old (1926, 1933, 1945, 1948, 1961)

2) 2.1 Plant material -  line 97 correct "stage. stage"

R=Addressed

Describe as stored the sample before analysis. (Temperature)

R=Addressed

2.2 extraction - the authors apply an extraction method very time and solvent consuming, a justification should be given on the choice of this method. In the lecterature, there are many other extraction method more appropriate.

R=The method was clarified and better described

2.4 Q-TOF MS/MS - the authors should give more informations. Ion selected for qualify and quantify. A description of the selction of the ions and the parameters of ion detection should be given in table.

R=The respective reference was added.

2.5 GC-MS analysis Information on the GC condition should be given (column type, oven condition, MS condition) The ions more adbundant used for the qualy-quanti analysis.

R=The reference including this information was added were corresponds.

References - The authors should implement the reference with more recent papers.

R=Addressed

In additon the authors should give information on the recoveries obtained with the applycation of method. (Fortification levels, number of replicates, acuracy, precision, linearity, LOD and LOQ)

R= Since the objective was the determination of the cardiac glycosides on mature and immature seeds, to characterize the compounds present in Thevetia thevetioides seeds (not previously described) we realized a qualitative analysis. However, we are working on the quantification and bioactivity of this compounds for future research, and we are considering this commentary. Thank you. 

Round 2

Reviewer 2 Report

Comments and Suggestions for Authors

The open point addressed. Now the paper is good.